**Data Availability Statement:** All relevant data are within the paper.

**Funding:** The author(s) received no specific funding for this work.

# Clinical correlates of sarcopenia and falls in Parkinson's disease

**Danielle Pessoa Lima**[1,2,3,4]*, **Samuel Brito de Almeida**[4], **Janine de Carvalho Bonfadini**[1,2,4], **João Rafael Gomes de Luna**[2], **Madeleine Sales de Alencar**[2,5], **Edilberto Barreira Pinheiro-Neto**[2], **Antonio Brazil Viana-Júnior**[4], **Samuel Ranieri Oliveira Veras**[1], **Manoel Alves Sobreira-Neto**[1,4,5], **Jarbas de Sá Roriz-Filho**[2], **Pedro Braga-Neto**[1,4,6]

**1** Division of Neurology, Department of Clinical Medicine, Universidade Federal do Ceará, Fortaleza, Brazil, **2** Division of Geriatrics, Department of Clinical Medicine, Universidade Federal do Ceará, Fortaleza, Brazil, **3** Medical School of Universidade de Fortaleza, Fortaleza, Brazil, **4** Clinical Research Unit of Hospital Universitário Walter Cantídio, Universidade Federal do Ceará, Fortaleza, Brazil, **5** Medical School of Universidade Unichristus, Fortaleza, Brazil, **6** Center of Health Sciences, Universidade Estadual do Ceará, Fortaleza, Brazil

* dra.daniellelima@gmail.com

## Abstract

### Background

Sarcopenia is a complex and multifactorial geriatric condition seen in several chronic degenerative diseases. This study aimed to screen for sarcopenia and fall risk in a sample of Parkinson's disease (PD) patients and to investigate demographic and clinical factors associated.

### Methods

This is a cross-sectional study. We evaluated 218 PD patients at the Movement Disorders Clinic in Fortaleza, Brazil, and collected clinical data including experiencing falls in the six months prior to their medical visit. Probable sarcopenia diagnosis was confirmed by using a sarcopenia screening tool (SARC-F questionnaire) and the presence of low muscle strength.

### Results

One hundred and twenty-one patients (55.5%) were screened positive for sarcopenia using the SARC-F and 103 (47.4%) met the criteria for probable sarcopenia.

Disease duration, modified Hoehn and Yahr stage, Schwab and England Activities of Daily Living Scale score, levodopa equivalent dose, probable sarcopenia and positive SARC-F screening were all associated with experiencing falls. Disease duration, lower quality of life and female gender were independently associated with sarcopenia. Experiencing falls was significantly more frequent among patients screened positive in the SARC-F compared to those screened negative.

**Competing interests:** The authors have declared that no competing interests exist.

## Conclusions

Sarcopenia and PD share common pathways and may affect each other's prognosis and patients' quality of life. Since sarcopenia is associated with lower quality of life and increased risk of falls, active case finding, diagnosis and proper management of sarcopenia in PD patients is essential.

## Introduction

Sarcopenia is a complex and multifactorial geriatric condition seen in several chronic degenerative diseases. It stems from abnormally reduced muscle mass quality and quantity and is associated to several negative outcomes such as falls, disability, poor quality of life, institutionalization, hospitalization, and death [1]. The prevalence of sarcopenia has increased and it has become a serious global public health concern [2]. Patients with Parkinson's disease (PD) show higher prevalence of sarcopenia and higher frequency of falls compared to non-PD patients [3,4]. Despite its importance, few studies have assessed the prevalence and characteristics of sarcopenia in this population [5–9].

Falls are a serious concern in PD with an annual incidence of 60% among PD patients. The risk of falls is greater in this population compared to healthy individuals and those with other neurological diseases with high risk of falls such polyneuropathy, spinal disorders and multiple sclerosis [10]. It is very important to early detect sarcopenia and fall risk in PD patients because they can benefit from simple interventions such as high-protein diet and resistance exercise training [5].

To the best of our knowledge, there are no large studies on the prevalence of sarcopenia and falls and associated risk factors in PD patients in Brazil. The aim of this study was to screen for sarcopenia and prevalence of falls in a sample of PD patients using a well-established screening tool for sarcopenia (SARC-F) [11]. In addition, we investigated demographic and clinical factors associated with sarcopenia and fall risk and their impact on the quality of life in PD patients.

## Materials and methods

### Study subjects

The study sample comprised consecutive patients with PD attending the Movement Disorders Clinic at Hospital Universitário Walter Cantídio in Fortaleza, Brazil, from January 2018 to August 31, 2019. The participants were regularly followed up at the clinic every 4–6 months. The diagnosis of PD was confirmed according to the Movement Disorders Society and the United Kingdom Parkinson's Disease Brain Bank criteria. Patients who did not meet the clinical diagnostic criteria for idiopathic PD were excluded. The study was approved by the Research Ethics Committee of Hospital Universitário Walter Cantídio and all participants gave their written informed consent (register number 91075318.1.0000.5045). All patients were interviewed and evaluated by the study investigators.

### Clinical evaluation

We used a structured interview to collect sociodemographic and clinical information including gender, age, age at onset of PD and disease duration. We evaluated past history of hypertension, diabetes, depression (according to the Diagnostic and Statistical Manual of Mental

Disorders, DSM-V) [12], dementia (according to DSM-V) [12] and osteoporosis (according to National Osteoporosis Foundation recommendations) [13]. Clinical information from the patients was cross-checked with data from relatives, caregivers, and clinical records for accuracy. We also collected information on antiparkinsonian drug treatments used including L-dopa (L-dopa/carbidopa, L-dopa/benserazide and controlled-release L-dopa formulations), COMT inhibitors (entacapone), MAO-B inhibitors (rasagiline), amantadine and dopamine agonists (pramipexole). We defined the levodopa equivalent dose (LED) of an antiparkinsonian drug as the dose that produces the same level of symptomatic control as 100 mg of immediate release L-dopa according to Tomlinson et al. systematic review [14]. We recorded the number of medications used by each patient and defined as polypharmacy if they were on 5 or more. We used the Schwab and England Activities of Daily Living (SE ADL) Scale [15] to evaluate ADL, the modified Hoehn and Yahr (HY) staging [16] to assess PD severity and the Parkinson's Disease Questionnaire (PDQ-39) [17] to assess quality of life. Depressive symptoms were assessed using the 15-item Geriatric Depression Scale (GDS-15) [18].

## Assessment of sarcopenia

The SARC-F was administered to all patients [11]. The European Working Group on Sarcopenia in Older People (EWGSOP) [19] recommends the use of the SARC-F as a way to elicit self-reports from patients on signs that are characteristic of sarcopenia and to introduce assessment and treatment of sarcopenia into clinical practice. The SARC-F is an inexpensive and convenient tool for screening sarcopenia risk that can be readily used in community healthcare units and other clinical settings. It has 5 items to assess the patient's perception of his or her limitations in strength, walking ability, rise from a chair, climb stairs and falls.

Functional lower extremity strength was measured with the Five Times Sit-to-Stand (FTSTS) test. Participants were asked to stand up from a sitting position and sit down 5 times as quickly as possible without pushing off. Those who could not rise with their arms folded across the chest were excluded from this test. Trained staff recorded the time a participant took to consecutively stand up 5 times from a seated position on a 45-cm-tall chair with the arms folded across the chest. The time from the command "Go" until the participant reached the chair on the fifth repetition was recorded once in seconds with a stopwatch. The cut-off point for low strength by FTSTS test is 15 seconds [20].

Hand grip strength was assessed using a dynamometer (JAMAR) for three repetitions in each hand. The average of three measures was recorded. The cut-off points for grip strength is 27 kg for men and 16 kg for women [21]. The highest value recorded in either hand was included in the statistical analysis.

The diagnosis of sarcopenia was based on EWGSOP criteria [19]. The diagnosis of probable sarcopenia was confirmed by screening positive in the SARC-F (score ≥4) and the presence of low muscle strength measured by the hand grip strength test and/or FTSTS. All patients were evaluated for disease staging (modified HY scale), quality of life, lower extremity strength (FTSTS) and hand grip strength during "on" phases.

## Assessment of falls

A fall was defined as an event where the patient inadvertently came to the ground or other lower level that was not a result of a violent behavior such as fight, car or bike accident, syncope or epilepsy. The patients were asked about experiencing such an event in the six months prior to their medical visit. Information of falls obtained from the patients was cross-checked with data from relatives, caregivers and clinical records for accuracy.

### Statistical analysis

For numerical variables, data were presented as means, standard deviations and medians. For categorical variables, data were described as frequencies and prevalence rates to investigate associations between risk factors and probable sarcopenia and falls. For the analysis of the participants' characteristics, we used the Mann-Whitney U-test because independent variables were not normally distributed. To investigate the association between categorical variables, we used Pearson's chi-square and Fisher's exact tests. Logistic regression models including probable sarcopenia and falls as outcomes were adjusted for the variables with $p < 0.05$ in the bivariate analysis.

A significance level of 5% was adopted. Statistical analyses were performed using the JAMOVI statistical software (Version 0.9).

## Results

### Associations with probable sarcopenia

The study population was composed of a total of 218 patients, of which 93 (42.7%) were women. The mean age was 67.2 ± 10.9 years and mean disease duration was 9.4 ± 6.9 years. One hundred and twenty-one patients (55.5%) screened positive for sarcopenia using the SARC-F and 103 (47.4%) met the criteria for probable sarcopenia. Among patients with probable sarcopenia, 61 (59.2%) and 87 (84.4%) had low muscle strength according to hand grip strength and FTSTS respectively. Table 1 shows clinical and sociodemographic characteristics of the study population and the results from the bivariate analysis for probable sarcopenia.

The following variables with $p < 0.05$ in the bivariate analysis were included in the logistic regression models: gender, falls, osteoporosis, depression, GDS score, modified HY stage, SE ADL score, LED, disease duration, number of medications and PDQ-39 score. These variables were included in the logistic regression models in a stepwise forward manner. Female gender, modified HY stage and PDQ-39 score were independently significantly associated with probable sarcopenia (Table 2).

### Associations with falls

Disease duration, modified HY stage, SE ADL score, LED, probable sarcopenia and positive SARC-F (SARC-F+) were associated with falls with $p < 0.05$ in the bivariate analysis (Table 3).

These variables were included in the stepwise forward logistic regression models. Table 4 shows that only SARC-F+ and disease duration were independently associated with falls in our study. The association between positive SARC-F and occurrence of falls is also illustrated in Fig 1.

## Discussion

In our study, we found a very high prevalence of sarcopenia in PD patients (47.4%). Recent systematic review and meta-analysis of population-based studies estimated an overall prevalence of sarcopenia of 10% in healthy adults aged ≥60 years [2] and a previous report of the International Sarcopenia Initiative have found a prevalence varying between 1% and 29% in community-dwelling populations aged ≥50 years [22].

There are possible explanations for this higher prevalence of sarcopenia in PD patients. Firstly, the decrease in the numbers of motoneurons is a common feature in sarcopenia and PD. The strength for movement is secondary to contraction induced by neural action potentials. Dopaminergic action that facilitates the movement is compromised in PD so that both sarcopenia and PD have overlapping pathophysiological mechanisms for muscle fiber loss:

**Table 1. Sociodemographic and clinical characteristics and results from the bivariate analysis for probable sarcopenia.**

| | | Probable sarcopenia | | |
|---|---|---|---|---|
| | Total | Yes | No | p |
| SARC-F+ a | 121 (55,5%) | 103 (100%) | 18 (15,7%) | <0,001d |
| Gender | | | | 0,013c |
| Female | 93 (42,7%) | 53 (51,5%) | 40 (34,8%) | |
| Male | 125 (57,3%) | 50 (48,5%) | 75 (65,2%) | |
| Age | 67.9 (59.8–75.6) | 70.4 (59.9–76.8) | 66.4 (59.4–73.6) | 0,116b |
| Disease duration | 7 (4–13) | 9 (5–14) | 6 (3–11) | 0,009b |
| Hoehn and Yahr stage | 2.5 (2–3) | 3 (2–3) | 2 (2–2.5) | <0,001b |
| SE ADS score | 80 (70–90) | 80 (60–90) | 90 (80–90) | <0,001b |
| LED | 1000 (600–1400) | 1125 (750–1449) | 900 (500–1398) | 0,029b |
| Number of medications | 4 (3–6) | 5 (3–7) | 4 (3–5) | 0,032b |
| Polypharmacy | | | | 0,151c |
| <5 | 117 (53,7%) | 50 (48,5%) | 67 (58,3%) | |
| ≥5 | 101 (46,3%) | 53 (51,5%) | 48 (41,7%) | |
| Hypertension | 91 (41,7%) | 43 (41,7%) | 48 (41,7%) | 0,999c |
| Type 2 DM | 42 (19,4%) | 23 (22,3%) | 19 (16,7%) | 0,292c |
| Dementia | 27 (12,4%) | 17 (16,5%) | 10 (8,7%) | 0,081c |
| Depression | 100 (46,1%) | 57 (55,3%) | 43 (37,7%) | 0,009c |
| Falls | 92 (43,8%) | 51 (51,5%) | 41 (36,9%) | 0,034c |
| GDS score | 6 (3–9) | 7 (4–10) | 4 (2–7) | <0,001b |
| Motor physical therapy | 24 (12%) | 13 (13,5%) | 11 (10,6%) | 0,519c |
| Osteoporosis | 21 (10%) | 16 (16%) | 5 (4,5%) | 0,006c |
| PDQ score | 40.4 (27.2–54.8) | 51.3 (38.5–62.2) | 31.4 (22.4–45.5) | <0,001b |
| Functional mobility | 47.5 (22.5–72.5) | 65 (42.5–80) | 30 (15–50) | <0,001b |
| Activities of daily living | 41.7 (25–62.5) | 54.2 (33.3–75) | 33.3 (16.7–45.8) | <0,001b |
| Emotional well-being | 41.7 (25–58.3) | 45.8 (33.3–66.7) | 37.5 (25–50) | 0,001b |
| Stigma | 31.3 (21.9–56.3) | 37.5 (25–68.8) | 25 (18.8–43.8) | 0,070b |
| Social support | 58.3 (41.7–66.7) | 58.3 (41.7–66.7) | 58.3 (41.7–66.7) | 0,832b |
| Cognition | 37.5 (21.9–50) | 43.8 (25–56.3) | 25 (12.5–50) | <0,001b |
| Communication | 33.3 (16.7–58.3) | 41.7 (25–58.3) | 25 (16.7–41.7) | 0,002b |
| Bodily discomfort | 50 (33.3–70.8) | 58.3 (41.7–83.3) | 50 (25–66.7) | 0,001b |
| Hand grip strength | 21.3 (15–30.6) | 18 (12–25) | 26.7 (18.8–34.3) | <0,001b |
| FTSTS | | | | <0,001b |
| Altered | 144 (66,1%) | 95 (92,2%) | 49 (42,6%) | |
| Not performed | 17 (7,8%) | 6 (5,8%) | 11 (9,6%) | |
| Normal | 57 (26,1%) | 2 (1,9%) | 55 (47,8%) | |

Data expressed in percentage (%), as well as mean ± standard deviation for normally distributed data and median (25th-75th) for not normally distributed

[a]: Sarcopenia was assessed using SARC-F scores, SARC-F +: score ≥4.

[b]: Mann-Whitney test

[c]: Pearson's chi-squared test

[d]: Fisher's Exact Test

SE ADL: Schwab and England Activities of Daily Living Scale; LED: levodopa equivalent dose; GDS: Geriatric Depression Scale; PDQ: Parkinson's Disease Questionnaire; Type 2 DM: Type 2 diabetes mellitus; FTSTS: Five Times Sit-to-Stand test.

inflammation, muscle autophagy, oxidative stress, and apoptosis [6]. Secondly, the loss of lean mass secondary to malnutrition is also prevalent in PD. PD patients are likely to experience nausea, dyspepsia, constipation, medication side effects, dysphagia, anorexia and depressive

**Table 2. Multivariate analysis for probable sarcopenia.**

| Predictors | OR (95% CI) | p-value |
|---|---|---|
| Female gender | 3.05 (1.34–6.94) | 0.008 |
| Hoehn and Yahr stage | 1.87 (1.06–3.29) | 0.03 |
| PDQ score | 1.06 (1.03–1.09) | <0.001 |

Stepwise-forward logistic regression models for probable sarcopenia. Variables included in the regression analysis: gender; falls; osteoporosis; Hoehn and Yahr stage; SE ADL: Schwab and England Activities of Daily Living Scale; LED: levodopa equivalent dose; disease duration; and PDQ score: Parkinson's Disease Questionnaire.

symptoms that cause reduced energy intake [23]. Thirdly, it has also been found that PD patients have lower levels of physical activity (in terms of amount and intensity) compared to healthy older adults [5,7,24].

To date, few studies have assessed the prevalence of sarcopenia in PD patients, with reports ranging from 6% to 55.8% [8,9,25,26]. It should be noted that the prevalence of sarcopenia varies greatly due to the use of different definitions and diagnostic tools as well as patient selection methods. However, these data are of great relevance given the increased risk of adverse outcomes associated with sarcopenia in the older adults such as falls, fractures and impaired ability to perform ADL [19], which are also important issues when managing PD patients.

Patients with probable sarcopenia and falls show higher modified HY and lower SE ADL scores. Moreover, they take higher LED and use a greater number of medications. It has been shown that older people with antiparkinsonian drugs are at higher risk of being exposed to fall-risk inducing drugs [27] The more advanced the disease, the more they require medications and the less independent these patients are [28]. As neurodegeneration progresses, body composition and physical performance are affected. Weight loss is very common in later PD stages and underweight patients show severe reductions in muscle quality and quantity [6].

Probable sarcopenia was significantly strongly associated with depression and higher GDS-15 scores. This association can be explained by physical inactivity, upregulation of inflammatory cytokines and dysregulation of hormones in the hypothalamic–pituitary–adrenal axis. A recent systematic review and meta-analysis of 10 observational studies concluded that sarcopenia was independently associated with depression [29].

Probable sarcopenia was associated with falls. Several studies have showed that reduced mobility, poor balance and reduced leg muscle strength have been associated with increased fall risk [30–31]. These signs are clinical manifestations of sarcopenia.

Disease duration, quality of life and female gender were independently associated with sarcopenia in this study. Evidence in the literature of factors associated with sarcopenia in PD patients is still scarce. In a recent cross-sectional study including 104 PD patients from a tertiary center in Innsbruck, Austria, sarcopenia was significantly associated with longer disease duration among other factors [9]. It suggests that, in addition to its high prevalence in this population, sarcopenia might be associated with factors related to disease progression. Reduced quality of life is a very important long-term adverse outcome. Furthermore, patients' quality of life was assessed in our study using the PDQ-39, which is a questionnaire that takes into account patient self-reports and observation. Thus, the association with reduced quality of life emphasizes that active case finding, diagnosis and proper management of sarcopenia in PD patients is essential. The association with female gender is controversial and it is not well established in the literature [9].

Falls in PD involve very complex multifactorial mechanisms and recurrent falls is a milestone of disease progression [32]. Falls were significantly more frequent in patients who were

**Table 3. Sociodemographic and clinical characteristics and results from the bivariate analysis for falls.**

| | Falls | | | |
|---|---|---|---|---|
| | **Total** | **Yes** | **No** | **p-value** |
| SARC-F+ [a] | 115 (54.8%) | 62 (67.4%) | 53 (44.9%) | <0.001[c] |
| Gender | | | | 0.518[c] |
| Female | 92 (43.8%) | 38 (41.3%) | 54 (45.8%) | |
| Male | 118 (56.2%) | 54 (58.7%) | 64 (54.2%) | |
| Age | 67.9 (59.8–75.6) | 69 (61.6–73.8) | 66.9 (57.8–76.8) | 0.632[b] |
| Disease duration | 7 (4–13) | 10 (5–17.5) | 6 (4–10) | <0.001[b] |
| Hoehn and Yahr stage | 2.5 (2–3) | 2.8 (2–3) | 2 (2–2.5) | 0.001[b] |
| SE ADL score | 80 (70–90) | 80 (70–90) | 90 (80–90) | 0.016[b] |
| LED | 1000 (600–1400) | 1125 (712.5–1524.5) | 900 (550–1350) | 0.024[b] |
| Number of medications | 4 (3–6) | 5 (3–7) | 4 (3–6) | 0.052[b] |
| Polypharmacy | | | | 0.065[c] |
| <5 | 111 (52.9%) | 42 (45.7%) | 69 (58.5%) | |
| ≥5 | 99 (47.1%) | 50 (54.3%) | 49 (41.5%) | |
| Hypertension | 87 (41.4%) | 33 (35.9%) | 54 (45.8%) | 0.149[c] |
| Type 2 DM | 41 (19.6%) | 20 (21.7%) | 21 (17.9%) | 0.493[c] |
| Dementia | 26 (12.4%) | 15 (16.3%) | 11 (9.3%) | 0.127[c] |
| Depression | 97 (46.4%) | 43 (47.3%) | 54 (45.8%) | 0.830[c] |
| Probable sarcopenia | 99 (47.1%) | 51 (55.4%) | 48 (40.7%) | 0.034[c] |
| Katz scale score | 2.3 ± 1.5 (2) | 2.4 ± 1.4 (2) | 2.3 ± 1.6 (2) | 0.446[b] |
| Pfeffer scale score | 9.6 ± 7.3 (8) | 9.6 ± 7.1 (9) | 9.6 ± 7.5 (8) | 0.989[b] |
| GDS score | 6 (3–9) | 6 (4–9) | 5 (3–8) | 0.573[b] |
| Motor physical therapy | 24 (12.3%) | 13 (15.5%) | 11 (9.9%) | 0.241[c] |
| Osteoporosis | 21 (10.4%) | 9 (10%) | 12 (10.7%) | 0.869[c] |
| PDQ score | 40.4 (27.2–55.1) | 43.6 (27.6–60.3) | 39.1 (26.9–51.3) | 0.149[b] |
| Functional mobility | 47.5 (22.5–72.5) | 46.3 (25–77.5) | 47.5 (22.5–70) | 0.532[b] |
| Activities of daily living | 41.7 (25–62.5) | 43.8 (25–70.8) | 37.5 (25–58.3) | 0.152[b] |
| Emotional well-being | 41.7 (25–58.3) | 41.7 (25–62.5) | 37.5 (25–58.3) | 0.340[b] |
| Stigma | 31.3 (18.8–56.3) | 31.3 (25–68.8) | 31.3 (12.5–50) | 0.169[b] |
| Social support | 58.3 (41.7–66.7) | 58.3 (41.7–66.7) | 58.3 (41.7–66.7) | 0.938[b] |
| Cognition | 37.5 (21.9–50) | 37.5 (25–56.3) | 37.5 (18.8–50) | 0.414[b] |
| Communication | 33.3 (16.7–58.3) | 33.3 (25–58.3) | 33.3 (16.7–58.3) | 0.362[b] |
| Bodily discomfort | 50 (33.3–75) | 54.2 (41.7–75) | 50 (33.3–70.8) | 0.296[b] |
| Hand grip strength | 21.7 (15–30.6) | 21.3 (14.7–28.7) | 22 (15.3–32.7) | 0.291[b] |
| FTSTS | | | | 0.067[c] |
| Altered | 141 (67.1%) | 62 (67.4%) | 79 (66.9%) | |
| Not performed | 16 (7.6%) | 11 (12%) | 5 (4.2%) | |
| Normal | 53 (25.2%) | 19 (20.7%) | 34 (28.8%) | |

Data expressed in percentage (%), as well as mean ± standard deviation for normally distributed data and median (25th-75th) for not normally distributed

a: Sarcopenia was assessed using SARC-F scores, SARC-F +: score ≥4.

b: Mann-Whitney test

c: Pearson's chi-squared test

d: Fisher's Exact Test; SE ADL: Schwab and England Activities of Daily Living Scale; LED: levodopa equivalent dose; GDS: Geriatric Depression Scale; PDQ: Parkinson's Disease Questionnaire; Type 2 DM: Type 2 diabetes mellitus; FTSTS: Five Times Sit-to-Stand test.

screened positive in the SARC-F compared to those screened negative. SARC-F is a very useful tool because it does not require special measurements or equipment as well as highly trained

**Table 4. Multivariate analysis for falls.**

| Predictors | OR (95% CI) | p-value |
|---|---|---|
| SARC-f+ | 1.87 (1.02–3.41) | 0.042 |
| Duration of disease | 1.10 (1.05–1.15) | <0.001 |

Stepwise-forward logistic regression models for falls. Variables included in regression: disease duration; Hoehn and Yahr stage; SE ADL: Schwab and England Activities of Daily Living Scale; LED: levodopa equivalent dose; probable sarcopenia; and SARC-F+

professionals for its administration. It is used to identify older adults with impaired physical function since it is a proven good predictive tool of physical limitations and mortality in community-dwelling older people [11]. The five items of SARC-F indirectly assess the risk of falls through muscle strength and balance. Tan et (2017) al. managed to identify 93% of patients at increased risk of adverse outcomes and care burden from falls using the SARC-F screening tool [33]. SARC-F is a questionnaire that is self-reported by the patient and reflects how much difficulty they face with ADL. Indeed, Almeida et al. (2015) identified self-reported disability as the strongest predictor of falls in a sample of 130 patients in a 12-month prospective study conducted in Bahia, Brazil [3].

Disease duration was associated with falls in the multivariate regression model in this study. Farombi et al. (2016) evaluated 81 PD patients and compared fallers versus non-fallers and showed that frequent fallers had significantly worse outcomes and longer disease duration [4]. PD symptoms are progressive and affect postural control in advanced stages and longer disease duration is associated with greater disease severity. Disease duration has also been correlated with the occurrence of falls in other studies [34].

The present study has some limitations. Firstly, we excluded patients who were wheelchair users. Therefore, the prevalence of sarcopenia may have been underestimated in our study population. Secondly, curves of normality and cut-off points for muscle mass and function to assess sarcopenia in PD are not available yet. In addition, low muscle mass was not measured by dual-energy X-ray absorptiometry (DXA) which is considered the gold standard for diagnosing this condition [35].

Thirdly, muscle strength testing in individuals with bradykinesia and rigidity is a subject of debate. Cano-de-la-cuerda et al (2010) [36] developed a systematic review with the objective of investigating whether muscle weakness exists in Parkinson's disease. Seventeen studies showed that the isokinetic muscle strength of patients with Parkinson's disease was reduced compared to age-matched controls. Interestingly, they concluded that muscle weakness was not specifically associated with tremor or stiffness. The specific etiology of this weakness requires further studies. It is necessary to establish better if this weakness is of central or peripheral origin, that is, if it is intrinsic to the disease or a secondary phenomenon. Duncan, Leddy and Earhart (2011) evaluated intra-rater and test-retest reliability of the FTSTS, the performance of patients in the test at different stages of the disease, its correlation to other measures and its ability to predict falls in 80 patients with PD [20]. They concluded that the test-retest reliability of the FTSTS in PD is excellent and it is comparable to that of other populations. They recommended the use of the FTSTS as a quick and objective measure to determine the risk of fall. The FTSTS was correlated to worse quality of live, to lower balance confidence, to lower limbs strength, to slower movements of upper extremity in bivariate analysis. The lower limbs strength was not correlated to FTSTS in multiple regression analysis. To the best of our knowledge, it was the first trial to examine FTSTS performance in individuals with PD. The sample size of the study was small so that we need more studies that examine the FTSTS in individuals with PD to

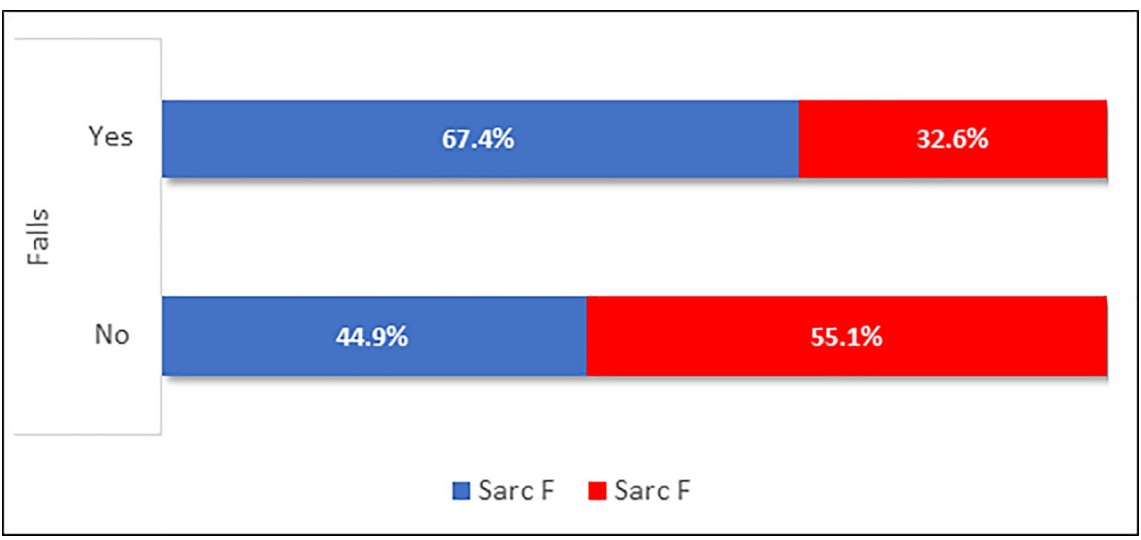

**Fig 1. Association between positive SARC-F and occurrence of falls.**

clarify its correlation with postural instability and bradykinesia. PD asymmetry should be considered, and bradykinesia, rigidity and tremor may be present unilaterally or bilaterally and one side is usually more affected. This is why we used greater hand grip strength, as suggested by Vetrano et al. (2018) [6], instead of grip strength of dominant hand, as recommended by the sarcopenia consensus [22]. Rising from a chair is a physically demanding function that can be compromised by muscle quality and PD symptoms.

Yet, our study has some strengths. The study's main finding of an increased prevalence of probable sarcopenia in PD suggests the presence of neurodegenerative processes leading to sarcopenia. We showed that a simple inexpensive tool (SARC-F) is associated with occurrence of falls. The value of SARC-F as a predictive tool should be further investigated in prospective studies. In addition, we highlighted the impact of probable sarcopenia on quality of life. Sarcopenia and falls pose an economic burden to our societies [22], yet an inexpensive intervention approach including high-protein diet and resistance exercise training can help reverse or improve this condition. The sooner these interventions are implemented, the more likely PD patients will benefit from them.

## Conclusions

Despite its clinical importance, sarcopenia diagnosis in PD patients has not been much explored in clinical practice. It is necessary to optimize the assessment of PD patients including body composition and mass and muscle strength assessments for the diagnosis of sarcopenia. Sarcopenia and PD share common pathways and, regardless of their source, they may affect each other's prognosis and patients' quality of life. The results of our study support the use of SARC-F for screening fall risk. We plan to further investigate sarcopenia and SARC-F as predictors of falls in a future prospective study.

## Acknowledgments

We gratefully acknowledge the patients that agreed to participate on this study.

## Author Contributions

**Conceptualization:** Danielle Pessoa Lima, Samuel Brito de Almeida, Pedro Braga-Neto.

**Data curation:** Janine de Carvalho Bonfadini, Madeleine Sales de Alencar, Edilberto Barreira Pinheiro-Neto.

**Formal analysis:** Antonio Brazil Viana-Júnior, Manoel Alves Sobreira-Neto.

**Investigation:** Danielle Pessoa Lima, Samuel Brito de Almeida, Janine de Carvalho Bonfadini.

**Methodology:** Danielle Pessoa Lima, Samuel Brito de Almeida, Janine de Carvalho Bonfadini, Samuel Ranieri Oliveira Veras, Jarbas de Sá Roriz-Filho, Pedro Braga-Neto.

**Supervision:** Manoel Alves Sobreira-Neto, Jarbas de Sá Roriz-Filho, Pedro Braga-Neto.

**Writing – original draft:** Danielle Pessoa Lima, João Rafael Gomes de Luna, Manoel Alves Sobreira-Neto, Jarbas de Sá Roriz-Filho.

**Writing – review & editing:** Danielle Pessoa Lima, Samuel Brito de Almeida, Janine de Carvalho Bonfadini, João Rafael Gomes de Luna, Pedro Braga-Neto.

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
