## [Decision Letter · Decision Letter 0]

20 Jan 2020

PONE-D-19-34534

Clinical correlates of sarcopenia and falls in Parkinson’s disease

PLOS ONE

Dear Dr. Lima,

Thank you for submitting your manuscript to PLOS ONE. After careful consideration, we feel that it has merit but does not fully meet PLOS ONE’s publication criteria as it currently stands. Therefore, we invite you to submit a revised version of the manuscript that addresses the points raised during the review process.

Major revisions are needed in the present form.  See the Reviewers' comments carefully and respond them appropriately.

We would appreciate receiving your revised manuscript by Mar 05 2020 11:59PM. To enhance the reproducibility of your results, we recommend that if applicable you deposit your laboratory protocols in protocols.io, where a protocol can be assigned its own identifier (DOI) such that it can be cited independently in the future. For instructions see: http://journals.plos.org/plosone/s/submission-guidelines#loc-laboratory-protocols

We look forward to receiving your revised manuscript.

Kind regards,

Masaki Mogi

Academic Editor

PLOS ONE

Journal Requirements:

2. Please state the full name of the ethics committee that approved this study in your methods section.

3. Please ensure that references have been provided for all the scales and questionnaires used in this study.

Reviewers' comments:

Reviewer's Responses to Questions

**Comments to the Author**

1. Is the manuscript technically sound, and do the data support the conclusions?

Reviewer #1: Yes

Reviewer #2: No

2. Has the statistical analysis been performed appropriately and rigorously? 

Reviewer #1: Yes

Reviewer #2: No

3. Have the authors made all data underlying the findings in their manuscript fully available?

Reviewer #1: Yes

Reviewer #2: Yes

4. Is the manuscript presented in an intelligible fashion and written in standard English?

Reviewer #1: Yes

Reviewer #2: Yes

5. Review Comments to the Author

Reviewer #1: I have read the tex with great interest. SARC-F works well in PD patients. Study is well organized. Clinical confaunders are comprehensively evaluated. I have some suggestions in the content of minör revision.

Abstract

Falls are mentioned in the method section as ‘’in the six months prior to their medical visit’’. This description is sufficient throughout the abstract section. Please remove this description after each ‘’falls’’ from result section.

Introduction

Line 68: Despite its importance, few studies have assessed the prevalence and characteristics of sarcopenia in this population (references?). Please add refrences.

Line 80: on the quality of life ‘’of’’ PD patients. Please write ‘’in’’ instead of ‘’of’’.

Materials and Metods

Please define how you diagnosed dementia and osteoporosis.

Line 137:’’ lower extremity strength (SARC-F) and strength during on phases.’’ Please change as ’’lower extremity strength (FTSTS) and hand grip strength during on phases.’’

Line 142: Information ‘’on the number’’ of falls. Please remove ‘’on the number’’ because you did not evaluate the recurrent falls.

Falls are mentioned in the method section as ‘’in the six months prior to their medical visit’’. This description is sufficient throughout the text. Please remove this description after each ‘’falls’’ from result and discussion sections.

Results

Line 164: Before defining Table 1 please give the number (%) of patients with low muscle strenght according to both hand grip strenght and FTSTS.

Line 190: Why did you not include depression or GDS scores and number of medications in logistic regression models because p <0.05 in these values? Also please define depression cut off point according to GDS scale in method section.

In Table 1 and 3 both median and mean±standart deviation are not necessary, please only give mean±standart for normally distributed data and give median (25th-75th) for not normally distributed.

In Table 1 and 3 total number of participants for each independent variable is not necessary. You can remove them. (Example: SARC-F+ a ‘’(n = 218)’’ is not necessary)

Table 4: Instead of ‘’time’’ please write ‘’duration’’ of disease

Discussion

Line 305: Disease duration was associated ‘’to’’ falls. Please write ‘’with’’ instead of ‘’to’’.

Line 307: ‘’compared falling versus non-falling’’ please change the statement as ‘’compared fallers versus non-fallers’’

Line 310: Disease duration has also been correlated ‘’to’’ the occurrence of falls in other studies. Please write ‘’with’’ instead of ‘’to’’.

References

There is a duplication of references. 6th and 15th are same.

Reviewer #2: Dear Authors

the main problem with the manuscript is regarding the diagnosis and definition of sarcopenia.

Using the time to rise from a chair in Parkinsonian patients the main limitation is that time is related to the brakinesia and not to the muscle function. Other tests should be used in this case for the diagnosis of sarcopenia

6. PLOS authors have the option to publish the peer review history of their article (what does this mean?). If published, this will include your full peer review and any attached files.

Reviewer #1: Yes: Firuzan Fırat Ozer

Reviewer #2: Yes: Fulvio Lauretani

---

## [Author Response · Author response to Decision Letter 0]

17 Feb 2020

Reviewer #1: 

I have read the text with great interest. SARC-F works well in PD patients. Study is well organized. Clinical confounders are comprehensively evaluated. I have some suggestions in the content of minor revision.

Response: We thank the reviewer for the comments. 

Abstract

Falls are mentioned in the method section as ‘’in the six months prior to their medical visit’’. This description is sufficient throughout the abstract section. Please remove this description after each ‘’falls’’ from result section. 

Response: The description was further removed throughout the abstract section, as suggested.

Introduction

Line 68: Despite its importance, few studies have assessed the prevalence and characteristics of sarcopenia in this population (references?). Please add references.

Response: References are now added to this sentence.

Line 80: on the quality of life ‘’of’’ PD patients. Please write ‘’in’’ instead of ‘’of’’.

Materials and Metods 

Response: We agree with the reviewer. The sentence was modified in accordance with the recommendations.

Please define how you diagnosed dementia and osteoporosis

Response: Dementia was diagnosed following the Diagnostic and Statistical Manual of Mental Disorders (DSM-V) criteria (called major neurocognitive disorder). The diagnostic assessment for osteoporosis was conducted according to National Osteoporosis Foundation recommendations. This information was included in Methods section.

Line 137:’’ lower extremity strength (SARC-F) and strength during on phases.’’ Please change as ’’lower extremity strength (FTSTS) and hand grip strength during on phases.’’

Line 142: Information ‘’on the number’’ of falls. Please remove ‘’on the number’’ because you did not evaluate the recurrent falls.

Falls are mentioned in the method section as ‘’in the six months prior to their medical visit’’. This description is sufficient throughout the text. Please remove this description after each ‘’falls’’ from result and discussion sections.

Response: We thank the reviewer for the suggestions. All modifications were performed according to the reviewer suggestions. 

Results

Line 164: Before defining Table 1 please give the number (%) of patients with low muscle strenght according to both hand grip strenght and FTSTS.

Response: Among patients with probable sarcopenia, 61 (59.2%) and 87 (84.4%) had low muscle strength according to hand grip strength and FTSTS respectively. We included this information in Results section.

Line 190: Why did you not include depression or GDS scores and number of medications in logistic regression models because p <0.05 in these values? 

Depression, GDS score and number of medications were in fact included in logistic regression models. This information is now included in Results section. 

Also please define depression cut off point according to GDS scale in method section.

Response: Although GDS was applied in this study, it was not used in order to establish a diagnosis of depression, which was defined according to DSM-V criteria. In this sense, a cut off point for GDS was not adopted in this study. 

In Table 1 and 3 both median and mean ± standart deviation are not necessary, please only give mean±standart for normally distributed data and give median (25th-75th) for not normally distributed.

In Table 1 and 3 total number of participants for each independent variable is not necessary. You can remove them. (Example: SARC-F+ a ‘’(n = 218)’’ is not necessary) 

Table 4: Instead of ‘’time’’ please write ‘’duration’’ of disease 

Discussion

Line 305: Disease duration was associated ‘’to’’ falls. Please write ‘’with’’ instead of ‘’to’’. 

Line 307: ‘’compared falling versus non-falling’’ please change the statement as ‘’compared fallers versus non-fallers’’ 

Line 310: Disease duration has also been correlated ‘’to’’ the occurrence of falls in other studies. Please write ‘’with’’ instead of ‘’to’’.

Response: We thank the reviewer for the suggestions. All modifications were performed according to the reviewer suggestions. 

References

There is a duplication of references. 6th and 15th are same. 

Response: We apologize the reviewer for this mistake. All References section was changed considering that and also because new references were added.

Reviewer #2: 

Dear Authors:

The main problem with the manuscript is regarding the diagnosis and definition of sarcopenia.

Using the time to rise from a chair in Parkinsonian patients the main limitation is that time is related to the bradykinesia and not to the muscle function. Other tests should be used in this case for the diagnosis of sarcopenia 

Response: The authors acknowledge the influence of bradykinesia on altered results in this test as a limitation of the study. The limitations regarding the Five Times Sit-to-Stand Test (FTSTS) are in the tenth paragraph of the discussion. Indeed, literature concerning diagnosis of sarcopenia in PD patients is extremely scarce. Nonetheless, the FTSTS is the only test recommended by the European Working Group on Sarcopenia in Older People (EWGSOP 2) to assess muscle strength of the lower limbs. In this sense, as a validated tool to assess probable sarcopenia due to reduced muscle strength in the lower limbs in PD patients is yet unavailable, and given the need to use a standardized approach, the authors decided to follow the most widely accepted recommendations. Moreover, in this study all patients were assessed during “on” phases in order to minimize this confounding factor. 

Cano-de-la-cuerda et al (2010) developed a systematic review with the objective of investigating whether muscle weakness exists in Parkinson's disease. Seventeen studies showed that the isokinetic muscle strength of patients with Parkinson's disease was reduced compared to age-matched controls. Interestingly, they concluded that muscle weakness was not specifically associated with tremor or stiffness. The specific etiology of this weakness requires further studies. It is necessary to establish better if this weakness is of central or peripheral origin, that is, if it is intrinsic to the disease or a secondary phenomenon. Duncan, Leddy and Earhart (2011) evaluated intra-rater and test-retest reliability of the FTSTS, the performance of patients in the test at different stages of the disease, its correlation to other measures and its ability to predict falls in 80 patients with PD. They concluded that the test-retest reliability of the FTSTS in PD is feasible and it is comparable to that of other populations. They recommended the use of the FTSTS as a quick and objective measure to determine the risk of fall. The FTSTS was correlated to worse quality of live, to lower balance confidence, to lower limbs strength, to slower movements of upper extremity in bivariate analysis. The lower limbs strength was not correlated to FTSTS in multiple regression analysis. To the best of our knowledge, it was the first trial to examine FTSTS performance in individuals with PD. The sample size of the study was small so that we need more studies that examine the FTSTS in individuals with PD to clarify its correlation with postural instability and bradykinesia.

References

Cano-de-la-Cuerda R, Pérez-de-Heredia M, Miangolarra-Page JC, Munoz-Hellín E, Fernández-de-las-Penas C. Is there muscular weakness in Parkinson's disease?. American Journal of Physical Medicine & Rehabilitation. 2010;89(1):70-6. doi: 10.1097/PHM.0b013e3181a9ed9b.

Duncan RP, Leddy AL, Earhart GM. Five times sit-to-stand test performance in Parkinson's disease. Arch Phys Med Rehabil. 2011;92(9):1431-6. doi: 10.1016/j.apmr.2011.04.008.

---

## [Decision Letter · Decision Letter 1]

26 Feb 2020

Clinical correlates of sarcopenia and falls in Parkinson’s disease

PONE-D-19-34534R1

Dear Dr. Lima,

We are pleased to inform you that your manuscript has been judged scientifically suitable for publication and will be formally accepted for publication once it complies with all outstanding technical requirements.

With kind regards,

Masaki Mogi

Academic Editor

PLOS ONE

Additional Editor Comments (optional):

No further comment.

Reviewers' comments:

Reviewer's Responses to Questions

**Comments to the Author**

1. If the authors have adequately addressed your comments raised in a previous round of review and you feel that this manuscript is now acceptable for publication, you may indicate that here to bypass the “Comments to the Author” section, enter your conflict of interest statement in the “Confidential to Editor” section, and submit your "Accept" recommendation.

Reviewer #1: (No Response)

Reviewer #2: All comments have been addressed

2. Is the manuscript technically sound, and do the data support the conclusions?

Reviewer #1: Yes

Reviewer #2: Yes

3. Has the statistical analysis been performed appropriately and rigorously? 

Reviewer #1: Yes

Reviewer #2: Yes

4. Have the authors made all data underlying the findings in their manuscript fully available?

Reviewer #1: Yes

Reviewer #2: Yes

5. Is the manuscript presented in an intelligible fashion and written in standard English?

Reviewer #1: Yes

Reviewer #2: Yes

6. Review Comments to the Author

Reviewer #1: (No Response)

Reviewer #2: The authors addressed all my comments. In particular, they reported the validity of using the time to reach from a chair in PD.

7. PLOS authors have the option to publish the peer review history of their article (what does this mean?). If published, this will include your full peer review and any attached files.

Reviewer #1: Yes: Firuzan Fırat Ozer

Reviewer #2: Yes: Fulvio Lauretani

---

## [Editor Report · Acceptance letter]

5 Mar 2020

PONE-D-19-34534R1 

Clinical correlates of sarcopenia and falls in Parkinson’s disease 

Dear Dr. Lima:

I am pleased to inform you that your manuscript has been deemed suitable for publication in PLOS ONE. Congratulations! Your manuscript is now with our production department. 

With kind regards,

on behalf of

Dr. Masaki Mogi 

Academic Editor

PLOS ONE